# Regulatory Role of Vacuolar Calcium Transport Proteins in Growth, Calcium Signaling, and Cellulase Production in *Trichoderma reesei*

**DOI:** 10.3390/jof10120853

**Published:** 2024-12-11

**Authors:** Letícia Harumi Oshiquiri, Lucas Matheus Soares Pereira, David Batista Maués, Elizabete Rosa Milani, Alinne Costa Silva, Luiz Felipe de Morais Costa de Jesus, Julio Alves Silva-Neto, Flávio Protásio Veras, Renato Graciano de Paula, Roberto Nascimento Silva

**Affiliations:** 1Department of Biochemistry and Immunology, Ribeirão Preto Medical School, University of São Paulo, Ribeirão Preto 14049-900, SP, Brazil; leticia_oshiquiri@hotmail.com (L.H.O.); pereiralucasmath@gmail.com (L.M.S.P.); dbmaues@gmail.com (D.B.M.); alinnecs09@hotmail.com (A.C.S.); lfmoraess21@gmail.com (L.F.d.M.C.d.J.); 2Department of Cellular and Molecular Biology and Pathogenic Bioagents, Ribeirão Preto Medical School, University of São Paulo, Ribeirão Preto 14049-900, SP, Brazil; beterosa@fmrp.usp.br; 3Department of Pharmacology, Ribeirão Preto Medical School, University of São Paulo, Ribeirão Preto 14049-900, SP, Brazil; julio.neto@duke.edu (J.A.S.-N.);; 4Department of Physiological Sciences, Health Sciences Centre, Federal University of Espirito Santo, Vitoria 29047-105, ES, Brazil; renato.paula@ufes.br; 5National Institute of Science and Technology in Human Pathogenic Fungi, Brazil

**Keywords:** *Trichoderma reesei*, calcium, transport, vacuole, cellulase

## Abstract

Recent research has revealed the calcium signaling significance in the production of cellulases in *Trichoderma reesei*. While vacuoles serve as the primary calcium storage within cells, the function of vacuolar calcium transporter proteins in this process remains unclear. In this study, we conducted a functional characterization of four vacuolar calcium transport proteins in *T. reesei*. This was accomplished by the construction of the four mutant strains ∆*trpmc1*, ∆*tryvc1*, ∆*tryvc3*, and ∆*tryvc4*. These mutants displayed enhanced growth when subjected to arabinose, xylitol, and xylose. Furthermore, the mutants ∆*trpmc1*, ∆*tryvc1*, and ∆*tryvc4* showed a reduction in growth under conditions of 100 mM MnCl_2_, implying their role in manganese resistance. Our enzymatic activity assays revealed a lack of the expected augmentation in cellulolytic activity that is typically seen in the parental strain following the introduction of calcium. This was mirrored in the expression patterns of the cellulase genes. The vacuolar calcium transport genes were also found to play a role in the expression of genes involved with the biosynthesis of secondary metabolites. In summary, our research highlights the crucial role of the vacuolar calcium transporters and, therefore, of the calcium signaling in orchestrating cellulase and hemicellulase expression, sugar utilization, and stress resistance in *T. reesei*.

## 1. Introduction

Each year, the global food industry discards over 500 million tons of plant-based organic waste, largely composed of cellulose, hemicellulose, and lignin, which form lignocellulosic biomass [1,2]. This has led to a growing interest in using these molecules to produce second-generation biofuels, which utilize non-edible substrates, unlike first-generation biofuel substrates [3,4]. However, the industrial-scale production of biofuels faces significant challenges. A key obstacle is the requirement for enzymes to catalyze the conversion of biomass to biofuel, contributing to about 48% of the minimum sales value in bioethanol production [5]. As such, future research should focus on devising cost-effective production methods to make the process economically feasible and contribute to the wider goal of sustainable energy production.

The filamentous fungus, *Trichoderma reesei*, is recognized for its capacity to produce enzymes that degrade lignocellulosic biomass, primarily cellulases and hemicellulases. The production of these enzymes is influenced by several factors. For instance, the substrate can function as an inducer (e.g., cellulose or lactose) or a repressor (e.g., glucose) [6]. Other influencing factors include medium conditions, such as pH, cultivation duration, and temperature. Additionally, the presence of metals, like zinc, manganese, strontium, and calcium ions [7,8,9,10,11], also plays a role, with the last being the central point of this study. Previous research has demonstrated that the supplementation of the culture medium with CaCl_2_ triggers the production of cellulases via a calcium-dependent signaling pathway [10]. This observation emphasizes the importance of understanding the role of calcium ions in enzyme production for biofuel applications.

In fungi, it has been established that external stress stimuli trigger the opening of calcium channels in the plasma membrane or intracellular compartments, facilitating the influx of calcium ions [12]. Three such ions interact with calmodulin (CAM), activating it and enabling it to bind to the A subunit of calcineurin (CNA1). Simultaneously, other calcium ions bind to the B subunit of calcineurin (CNB1), thereby activating it. Calcineurin, a serine-threonine phosphatase, is composed of the catalytic subunit CNA1 and the regulatory subunit CNB1. CNA1 is known to interact with a variety of proteins, including transcription factors and other enzymes. Among these, the transcription factor CRZ1 (calcineurin-responsive zinc-finger transcription factor 1) is a notable example. Upon interaction, CNA1 dephosphorylates CRZ1, which consequently is translocated to the cell nucleus via Nmd5, which permits it to bind to CDREs (calcineurin-dependent response elements) in target promoters. This binding event induces the production of enzymes, such as cellulases and calcium transport proteins, including PMC1 and PMR1. This process ultimately leads to a reduction in the cytoplasmic calcium concentration until it reaches its basal level [13,14]. In fungi, vacuoles serve as the primary storage sites for calcium. As such, the transport system plays a pivotal role in maintaining calcium homeostasis and ensuring proper signal transduction. Fungi possess three types of vacuolar calcium transport proteins. In the case of *Saccharomyces cerevisiae*, calcium is released from the vacuoles via the Ca^2+^ channels YVC1 (or TRPY1). Following a peak in calcium levels, the Ca^2+^ ATPase PMC1 sequesters the calcium, while the Ca^2+^/H+ antiporter VCX1 maintains the basal level of calcium [12,15].

In *T. reesei*, numerous genes have been identified that encode proteins involved in calcium signaling. These include 3 ion channels, 10 ATPases, 10 transporters, 5 phospholipases C (PLC), 1 CAM, 1 CNA1, 1 CNB1, 9 genes that interact with calcium or CAM, and 8 additional genes involved in calcium-dependent signaling [16]. Our research group has observed the modulation of the expression of these calcium-related genes during the cultivation of *T. reesei* in the presence of various substrates, such as cellulose, sophorose, glucose, sugarcane bagasse, and glycerol. Notably, this modulation includes calcium transport proteins [6,17,18,19,20]. Furthermore, it has been demonstrated that the transcription factor CRZ1 plays a crucial role in the induction of the transcription factor XYR1, which subsequently leads to the production of cellulases, hemicellulases, and calcium and sugar transporters. The presence of calcium is a decisive factor for this process to occur [11].

Other studies have explored the influence of calcium on *T. reesei*. In 2016, Chen et al. [10] established that the transcription factor CRZ1 competes with the ACE1 repressor for the *xyr1* promoter, leading to the induction of cellulases. Subsequently, in 2018, Chen et al. [7] demonstrated that an elevated Mn^2+^ concentration results in an increase in intracellular Ca^2+^ levels via the P-type ATPase, TPMR1. This increase triggers the induction of genes encoding cellulases through the calcium-dependent signaling pathway. In 2019, Chen et al. [21] found that the addition of N, N-dimethylformamide (DMF) also stimulates the production of cellulases, an effect linked to the activation of PLC-E. In 2021, the mechanism underlying the production of cellulases from Mn^2+^ and DMF was explored in greater detail. Chen et al. [22] showed that both substances augment the amount of cyclic adenosine 3′,5′-monophosphate (cAMP) in the cell cytosol, leading to an increase in intracellular calcium concentration, a process dependent on PLC-E. In 2022, Li et al. [9] revealed that the addition of Sr^2+^ to the culture medium also causes an increase in intracellular calcium concentration, activating CRZ1 and resulting in enhanced cellulase activity. In 2023, Li et al. [8] demonstrated that Zn^2+^ boosts the activity of cellulases and xylanases, with PLC-E playing a role in this process by promoting the release of calcium from intracellular stores to the cytoplasm and by regulating the transcription factor ZAFA, which is involved in zinc metabolism and may influence cellulase production. Most recently, a study by Liu et al. [23] explored the effect of polyethylene glycol 8000 (PEG 8000) stress on cellulase biosynthesis in *T. reesei* via calcium signaling, providing further evidence of the crucial role of calcium in cellulase production. From these studies, it is evident that multiple factors can stimulate the production of cellulases via the calcium-dependent signaling pathway.

In this study, we undertook a comprehensive characterization of four vacuolar calcium transport proteins in *T. reesei* by deleting the genes encoding these proteins, leading to the generation of the mutant strains ∆*trpmc1*, ∆*tryvc1*, ∆*tryvc3*, and ∆*tryvc4*. These mutants exhibited enhanced growth when arabinose, xylitol, and xylose were used as a sole carbon source. Moreover, the mutants ∆*trpmc1*, ∆*tryvc1*, and ∆*tryvc4* displayed diminished growth in the presence of 100 mM MnCl_2_, suggesting a role in manganese resistance. Our enzymatic activity assays indicated the absence of the usual increase in cellulolytic activity observed in the parental strain upon the addition of calcium. The genes also influenced secondary metabolites’ expression, and confocal microscopy suggested calcium retention within the vacuoles in the absence of these transport proteins. These findings contribute to our understanding of calcium’s role in *T. reesei*’s growth and enzyme production.

## 2. Materials and Methods

### 2.1. Strains and Culture Conditions

The *T. reesei* strain QM6aΔ*tmus53*Δ*pyr4*, characterized by a deficiency in the nonhomologous end joining (NHEJ) repair pathway and auxotrophy for uridine, is herein designated as the parental strain. This strain was obtained from the Technical University of Vienna, Austria [24], and for its cultivation, 5 mM of uridine was added due to the deletion of the *pyr4* gene. The strains QM6aΔ*tmus53*Δ*58952*, QM6aΔ*tmus53*Δ*74057,* QM6aΔ*tmus53*Δ*55731*, and QM6aΔ*tmus53*Δ*56440* (hereafter referred to as Δ*trpmc1*, Δ*tryvc1*, Δ*tryvc3* and Δ*tryvc4* respectively) were generated during this study. These strains are distinguished by the deletion of calcium transport genes. For experimental procedures, 2 µL of a 10^7^ spore suspension of these strains in 0.08% NaCl and 0.05% Tween 80 were inoculated on MEX medium plates (3% malt extract (*w*/*v*) and 2% bacteriological agar (*w*/*v*)). The plates were subsequently incubated at 30 °C for 7–12 days.

To conduct the growth experiments, 10^7^ conidia in spore suspension were inoculated in minimal medium [25] agar plates containing 1% carboxymethylcellulose (CMC) or 2% glucose + 100 mM Congo red (CR). These plates were incubated for 3–5 days, and their growth was recorded daily. The microplate assays were conducted in minimal medium with the addition of 25 mM of cellobiose, galactose, glycerol, glucose, lactose, maltose, mannose, or xylose to obtain growth profiles in carbon sources, or with the addition of 0; 2.7; 5; 10; 15; 50; 100 or 150 mM of CaCl_2_, accompanied by 2% glucose to verify calcium tolerance. In addition, 0.25% phytagel and 0.03 Tween 20 was added to the medium. In these experiments, the microplates were incubated at 30 °C for 96 h. Absorbance readings at 750 nm were performed every 24 h to determine the degree of turbidity, as described in [26], indicating fungal growth. The experiments were carried out in three technical replicates and at least two biological replicates.

To perform the enzymatic assays and the gene expression quantification, we cultured the strains in Mandels Andreotti (MA) medium [27] with 24 h pre-growth in 1% glycerol followed by a transference of the mycelia to fresh MA media containing 1% sugarcane bagasse or 1% cellulose and incubated it for up to 72 h at 30 °C. Alternatively, we cultured the strains directly in 1% cellulose, 25 mM xylose, or 2% glucose for up to 72 h at 30 °C. Right after, the mycelia were filtrated, frozen using liquid nitrogen, and stored at −80 °C until use. The supernatant was collected, centrifuged for 10 min at 12,000× *g,* and frozen at −20 °C until use.

### 2.2. Vector Construction for Gene Deletion

The knockout strains were designed using the orotidine-5′-phosphate decarboxylase gene of *T. reesei* (*pyr4*, *Tr_74020*) as a selection marker, which enables the identification of transformants through auxotrophic complementation of the *T. reesei* parental strain. The full sequences of the genes cited in this study can be searched in the *Trichoderma reesei* v2.0 genome available in JGI database (https://mycocosm.jgi.doe.gov/Trire2/Trire2.home.html, accessed on 30 August 2024), except for *74057* (*tryvc1*), which the full coding region annotation can be found in *Trichoderma reesei* QM6a genome (https://mycocosm.jgi.doe.gov/Trire_Chr/Trire_Chr.home.html, accessed on 30 August 2024).

To facilitate genomic integration via homologous recombination, we constructed the cassettes, including regions with homology to approximately 1000 base pairs (bp) upstream and downstream of the target genes in *T. reesei* parental strain. For this purpose, two strategies were used. First, to construct the plasmids containing the cassettes of the genes *tryvc1* and *tryvc2*, primers for the amplification of its upstream and downstream regions were designed with the addition of restriction sites, allowing for ligation into the pJET-*pyr4* plasmid [28] (Appendix A), which already contains the sequence of the *pyr4* gene. The enzyme T4 DNA Ligase (Thermo Fisher Scientific, Waltham, MA, USA) was used to join the fragments, according to the manufacturer’s instructions. For the transformation of the ligation reactions, the heat-shock method was used in thermocompetent cells of *Escherichia coli* strain DH5α, according to [29]. After bacterial growth, plasmid DNA was extracted, as in [30], and digested with the respective restriction enzymes to confirm the ligation. In this strategy, the upstream region was first ligated, followed by the downstream region.

To construct the deletion cassettes of the genes *tryvc3* and *trpmc1*, primers were designed for the amplification of its upstream and downstream regions (Appendix A), as well as for the coding region of the *pyr4* gene. These primers included sequences of approximately 20 bp overlap between the fragments to be joined and the pRS426 plasmid (Appendix A), allowing the use of a yeast-mediated homologous recombination technique. In this case, the yeast shuttle vector pRS426 (*amp*^R^ *lacZ* URA3) [31] was digested with *Eco*RI and *Xho*I (Thermo Scientific, Waltham, MA, USA) and was purified with the QIAquick PCR Purification Kit (Qiagen, San Diego, CA, USA). Yeast transformation was essentially conducted, as previously described [32,33,34]. Here, an overnight culture (200 rpm, 30 °C) of the yeast *S. cerevisiae* strain SC9721 (*MATα his3-*Δ*200 URA3-52 leu2*Δ*1 lys2*Δ*202 trp1*Δ*63*) (Fungal Genetic Stock Center) was prepared. Then, 1 mL of the overnight culture was added to 50 mL of fresh YPD (1% yeast extract, 2% peptone, 1% glucose) (all from Sigma Aldrich, St. Louis, MO, USA) medium and incubated at 30 °C until OD_600_ = 1. Thereafter, the cells were centrifuged and resuspended in 100 mM lithium acetate for transformation. To accomplish this, we mixed equal amounts of the 5′ and 3′ flanking regions, *pyr4*, and the digested pRS426, and then used this mixture for yeast transformation using the lithium acetate method [32]. *S. cerevisiae* SC9721 transformants were selected for their ability to grow on YPD medium supplemented with lysine, histidine, leucine, and tryptophan, without uracil. After DNA extraction [35], the cassettes amplified by PCR were used for transformation in *T. reesei* parental strain. The deletion cassettes of *tryvc3* and *trpmc1* were PCR-amplified using 5F and 3R primers (Appendix A) [36]. All amplifications were performed from *T. reesei* parental strain genomic DNA, isolated according to [37]. After amplification, the fragments were purified using the QIAquick PCR Purification Kit (Qiagen). The sequences of the primers are listed in Appendix A.

### 2.3. Transformation of T. reesei

*T. reesei* QM6aΔ*tmus53*Δ*pyr4* was used for transformation to achieve highly efficient homologous integration of the deletion cassette. Protoplast transformation was carried out as previously described [24]. For the transformation of the *T. reesei* parental strain, approximately 10–40 µg of the linearized plasmid or cassette (amplified by PCR) was used for protoplast transformation [24].

Transformants were grown on selective minimal medium [1 g/L MgSO_4_·7H_2_O, 10 g/L 1% KH_2_PO_4_, 6 g/L (NH_4_)_2_SO_4_, 3 g/L trisodium citrate·2H_2_O, 10 g/L glucose, 20 mL/L 50× trace elements solution (0.25 g/L FeSO_4_·7H_2_O, 0.07 g/L ZnSO_4_·2H_2_O, 0.1 g/L CoCl_2_·6H_2_O, 0.085 g/L MnSO_4_·H_2_O), 2% (*w*/*v*) agar lacking uridine] (all from Sigma Aldrich). Additionally, the transformants underwent 3 rounds in Mandels-Andreotti medium [27] + 2% glucose without uridine with the addition of 0.1% (*v*/*v*) Triton X-100 for homokaryons selection Finally, to confirm the obtaining of the mutant strains, we used conventional PCR and RT-qPCR, as illustrated in Appendix A. The sequences of the primers are listed in Appendix A.

### 2.4. Enzyme Activity

Filter paper activity (FPase) was determined by an enzymatic reaction employing Whatman filter paper no. 1, 30 µL of culture supernatant, and 30 µL of 100 mM citrate-phosphate buffer (pH 5.0). The reactions were incubated at 50 °C for 20 h. Subsequently, 60 µL of 3,5-dinitrosalicylic acid (DNS) was added to the reaction, which was then heated at 95 °C for 5 min. The FPase activity was assayed in a 96-well microplate, and absorbance was read at 540 nm. Additionally, CMCase (Endoglucanase activity), β-glucosidase, xylanase, and β-xylosidase were assessed following the method described by [38] with minor modifications [17,39,40]. The CMCase activity was determined using 30 μL of 1% CMC (Sigma Aldrich, St. Louis, MO, USA) in sodium acetate buffer (50 mM, pH 4.8) and 30 µL of the enzyme (culture supernatant) at 50 °C for 1 h. Right after, 60 μL of DNS was added, followed by heating at 95 °C for 5 min to allow color development. Then, the absorbance of the samples was read at 540 nm.

To determine the β-glucosidase and β-xylosidase activities, p-nitrophenyl-derived substrates: p-Nitrophenol-β-D-glucopyranoside (pNP-Gluc) (5 mM) and p-Nitrophenyl-β-D-xylopyranoside (pNP-Xyl) (5 mM) were used, respectively. The β-glucosidase and β-xylosidase reactions were carried out in a microplate assay format and the assay mixture contained 10 μL of enzyme solution (culture supernatant), 50 μL of 50 mM sodium acetate buffer, and 40 μL of p-nitrophenyl-derived solution. The mixtures were buffered at pH 5.5 (β-glucosidase) and pH 4.8 (β-xylosidase) and after 15 min at 50 °C the reactions were stopped by adding 100 μL of 1 M sodium carbonate. The amount of p-nitrophenol was determined using a spectrophotometer at 405 nm.

Finally, xylanase activity was determined by mixing 25 µL of 1% xylan beechwood (Sigma Aldrich, St. Louis, MO, USA) in sodium acetate buffer (100 mM, pH 5.0) and 10 µL of enzyme and incubating at 50 °C for 30 min. The reaction was stopped by adding 75 µL of DNS, and the samples were heated for 5 min at 95 °C, followed by measuring the absorbance at 540 nm. In all enzymatic assays, one enzyme unit was defined as the amount of enzyme capable of liberating 1 µmol of reducing sugar or p-nitrophenol per minute per mL of sample solution (U/mL) [41]. All enzyme assays were completed in triplicate for each sample.

### 2.5. RNA Extraction and Transcript Analysis Using Quantitative PCR (RT-qPCR)

Total RNA was isolated from the mycelia of *T. reesei* strains using TRIZOL^®^ reagent (Thermo Fisher Scientific, Waltham, MA, USA) according to the manufacturer’s instructions. The quantification of RNAs was carried out by OD 260/280 spectrophotometry and their integrity was verified using the Agilent 2100 Bioanalyzer and electrophoresis in 1% agarose gel. Right after, 1 µg of total RNA was treated with DNase I (Thermo Fisher Scientific, Waltham, MA, USA) to eliminate genomic DNA contamination and used for cDNA synthesis using Maxima™ First Strand cDNA Synthesis kit (Thermo Fisher Scientific, Waltham, MA, USA) according to the manufacturer’s instructions. Then, cDNA was diluted 1:50 and analyzed using the CFX96™ Real-Time PCR Detection System (Bio-Rad Laboratories, Hercules, CA, USA) and SsoFast™EvaGreen^®^ Supermix (Bio-Rad Laboratories, Hercules, CA, USA), in accordance with the manufacturer’s instructions. Each reaction (10 µL) contained 5 µL of SsoFast™ EvaGreen^®^ Supermix (Bio-Rad Laboratories, Hercules, CA, USA), forward and reverse primers (300 nm each; Appendix A), cDNA template, and nuclease-free water. PCR cycling conditions were as follows: 10 min at 95 °C, followed by 40 cycles of 10 s at 95 °C and 30 s at 60 °C. Melt analysis used a ramp of 60–95 °C at a rate of 0.5 °C/10 s to evaluate primer dimers and nonspecific amplification. The β-actin gene was used as an endogenous control to normalize the total amount of cDNA present in each reaction [17,42].

### 2.6. Confocal Microscopy

To detect intracellular calcium, the Fluo4-AM reagent (Thermo Fisher Scientific, Waltham, MA, USA) was used. Firstly, 10^7^ conidia from each strain were incubated in MA medium for 48 h at 200 rpm and 30 °C in the presence or absence of calcium. After incubation, the cells were washed three times with PBS and loaded with 5 µM Fluo4-AM for 30 min at 30 °C. Then, the fluorescence intensity of Fluo4-AM was measured by excitation at a wavelength of 340 nm with an emission of 550 nm.

### 2.7. Bioinformatics and Statistical Analysis

To identify the calcium transporters and channels in *T. reesei* vacuoles, a set of homologous proteins previously characterized in other fungi were selected as references (Appendix A). Using the amino acid sequences of these proteins, three alignments with MAFFT were performed [43] and three profiles with hmmbuild were created [44]: (a) YVC1, (b) VCX1, and (c) PMC1. Then, these profiles were used to search the *T. reesei* proteome [45] using hmmsearch [44], and annotations were made using InterProScan [36]. For the cladogram design, the identified sequences and sequences of characterized proteins presented in [12] were employed. Finally, the sequences were aligned using MAFFT [43], cladogram was inferred using FastTree 2 [46] and displayed by iTOL v5 [47].

To analyze the gene expression of the identified proteins in the RNA-Seq assays conducted by our research group [6,17,18,19,20,48], were used a cut-off of 1.0 × 10^−45^ for the full e-value. We then used the gene expression levels under various conditions to create heatmaps using the python library seaborn [49].

All the results in this study were analyzed and statistically compared using the GraphPad Prism version 8.0 for Windows (www.graphpad.com, accessed on 30 August 2024). Student’s *t*-test was used for comparisons between the two groups. For comparisons between multiple groups, two-way ANOVA followed by Tukey’s multiple comparisons test was employed. All statistical tests assumed a 95% confidence interval.

## 3. Results

### 3.1. Identification of Putative Homologues in T. reesei: Three for PMC1, Five for YVC1, and Seven for VCX1 of S. cerevisiae

To identify calcium transporters and channels in *T. reesei* vacuoles, we used a set of homologs, previously characterized in other fungi, as references (Appendix A). We performed an alignment for each protein type—YVC1, VCX1, and PMC1 and used these alignments to create sequence profiles. These profiles were then applied to search the *T. reesei* proteome. This search led to the identification of a significant number of proteins, totaling 33 (Table 1). However, it is important to note that, while these proteins were identified, not all may be involved in calcium transport within the vacuoles.

To further investigate this hypothesis, we constructed a cladogram using the sequences of characterized calcium-transporting proteins located in various cellular compartments, as discussed in [12], and the proteins identified in this study (Figure 1). In terms of vacuolar proteins, we identified three proteins of *T. reesei* (75347, 62362 and 58952) that grouped with PMC1 homologs, five proteins (56440, 55731, 71037, 63125 and 740257) that grouped with YVC1 homologs and seven proteins (68169, 79599, 56744, 79398, 55595, 82544 and 62835) that grouped with VCX1 homologs. Other proteins initially identified as potential PMC1 homologs were found to be closest in the cladogram to ECA1/NCA-1, PMR1, SPF1, or isolated (unknown). In fact, the annotation in Table 1 indicated only three calcium-translocating P-type ATPases of the PMCA-type, but it is likely that other proteins were identified in our search due to their similarities with proteins localized in the endoplasmic reticulum and the Golgi apparatus. The number of calcium transport proteins in *T. reesei* underscores the complexity of the calcium flow control system in this organism.

Next, we analyzed the expression profiles of the predicted *T. reesei* vacuolar transport proteins using RNA-Seq data from our research group across a range of carbon sources and strains [6,17,18,19,20,48]. Our analysis specifically targeted the homologs of YVC1 and PMC1.

For the genes encoding YVC1 homologs (Figure 2a), we noted that in the commonly used parental strains QM9414, QM6a, and TU6, the genes *55731*, *74057*, and *56440* were notably upregulated under the conditions analyzed, particularly those that induce the production of cellulolytic enzymes, such as cellulose compared to glucose. In the mutant strains, we observed the downregulation of genes *71037*, *55731*, and *63125* in the absence of XYR1 in cellulose and sophorose, and of TMK2 in sugarcane bagasse. We also noted the upregulation of genes *74057* and *71037* in the absence of AZF1 in sugarcane bagasse and TMK2 in both sugarcane bagasse and glucose. Interestingly, all genes homologous to YVC1 were upregulated in the absence of CRE1 in sophorose, except for *63125*.

In relation to the genes homologous to PMC1 (Figure 2c), we primarily observed that gene *58952* is upregulated in QM6a under both conditions analyzed. Interestingly, we found that gene *58952* was differentially expressed only in the absence of AZF1. The genes *75347* and *62362* were slightly upregulated in the commonly used parental strains in all conditions, except in QM9414 in the presence of sophorose in relation to cellulose. Finally, gene *75347* was also downregulated in the mutant strains lacking CRE1, TMK1, and TMK2, while gene *62362* was downregulated only in the absence of TMK2.

Based on the RNA-Seq results from our research group and existing annotations in the JGI database, the genes *74057*, *55371*, *56440*, and *58952* were selected for further investigation in this study. The proteins encoded by these genes were designated as TrYVC1, TrYVC3, TrYVC4, and TrPMC1 respectively, based on the data obtained in the cladogram (Figure 1).

### 3.2. The Vacuolar Calcium Transport Proteins Are Involved in Sugars Assimilation, Manganese and Osmotic Stress Responses, Cellulose Deconstruction, and Cell Wall Stress Resistance

To elucidate the roles of TrYVC1, TrYVC2, TrYVC3, and TrPMC1 proteins of *T. reesei*, we engineered mutant strains lacking these genes (Appendix A). Then, we conducted a comprehensive evaluation of the effects of these gene deletions on growth under various conditions, such as different carbon sources with or without calcium supplementation, varying metal concentrations, and other stress factors. 

In our experiments, we monitored growth in minimal medium over a period of 72 h, using the carbon sources arabinose, cellobiose, galactose, glucose, glycerol, lactose, maltose, mannose, xylitol, and xylose (Figure 3 and Appendix A). On the first day, all mutant strains, except ∆*tryvc4* with CaCl_2_, exhibited enhanced growth compared to the parental strain when grown in the presence of arabinose, xylitol, and xylose (Appendix A). However, by the third day, calcium supplementation, which promoted growth in the parental strain, either inhibited growth or had no effect on the mutant strains (Figure 3). In the case of cellobiose, we observed a decline in the growth of mutant strains compared to the parental strain, particularly at 72 h (Figure 3). For maltose, all strains showed improved growth with calcium supplementation (Figure 3). These results suggest that these proteins are important to sugar assimilation and calcium supplementation might increase sugar uptake or metabolism in the absence of functional vacuolar calcium transport. However, these effects are carbon source dependent.

We further analyzed the growth of the strains in the presence of different metals and concentrations over a period of 48 or 72 h, using xylose as the carbon source (Figure 4). This approach was based on our observation that the mutant strains were more impacted by calcium supplementation in arabinose, xylitol, and xylose (Figure 3), and considering the cost-effectiveness of xylose. The parental strain demonstrated increased growth in the presence of 10 mM of all metals, except for cobalt. In higher concentrations, growth persisted only in the presence of calcium, manganese, and magnesium (Figure 4a,b,f,m). Interestingly, in contrast to the parental strain, the mutant strains displayed heightened sensitivity to 100 mM manganese (Figure 4e,f) and 10 mM zinc (Figure 4i,j), suggesting their importance in conferring resistance to these metals. Moreover, the growth enhancement typically induced by calcium in the parental strain was not observed in the mutant strains cultured in the absence of calcium (Figure 4a,b).

We examined the tolerance of *T. reesei* parental and mutant strains to elevated osmotic pressure by growing them on plates under osmotic stress conditions, specifically in the presence of 0.5 M NaCl or 0.5 M KCl, with and without calcium (Figure 5). Both NaCl and KCl inhibited growth of all strains at 24 h (Figure 5a,d), but by 72 h, the strains recovered and exhibited growth levels comparable to or higher than the control condition (Figure 5c,f). Notably, calcium supplementation led to a reduction in growth in the mutant strains, but not in the parental strain, at both 24 and 48 h (Figure 5a,b,d,e). This observation suggests that effective calcium signaling plays a pivotal role in *T. reesei*’s initial response to osmotic stresses.

Furthermore, the integrity of *T. reesei* parental e mutant strain’s cell wall was investigated by testing of sensibility to cell wall interfering substance Congo Red (CR). For this, we monitored growth in MA media with 1% CMC or 2% glucose + 100 mM Congo Red (CR) (Figure 6a). Interestingly, all mutant strains showed reduced growth at 1% CMC (Figure 6b), suggesting a role for vacuolar transport proteins in cellulose metabolism. Moreover, the strains ∆*trpmc1*, ∆*tryvc1*, and ∆*tryvc3* demonstrated enhanced growth on glucose in the presence of the cell wall stressor CR (Figure 6c). Given that the mutant strains exhibit reduced growth in glucose at the 72-h mark (Figure 3), it is plausible that the observed growth enhancement in the presence of CR may be attributed to the impact of the absence of vacuolar calcium transport proteins on the cell wall, suggesting the knockout strains have a high tolerance to stress agents such as CR.

### 3.3. TrPMC1, TrYVC1, TrYVC3, and TrYVC4 Are Key Factors in Cellulases Production Under Calcium Supplementation

We have shown that the strains lacking the proteins TrPMC1, TrYVC1, TrYVC3, and TrYVC4 exhibit impaired growth when CMC is used as the carbon source in the media (Figure 6b). To assess the impact of deleting genes encoding vacuolar calcium transport proteins on cellulase production, we cultivated both the parental and the mutant strains in Mandels-Andreotti (MA) medium with 1% cellulose, both in the presence or absence of 10 mM CaCl_2_.

Regarding CMCase activity we noted an increase in the parental strain supplemented with calcium at all analyzed time points (Figure 7a) being this increase also observed in ∆*trpmc1* and ∆*tryvc1* at 48 h of cultivation in the presence of calcium, albeit at a significantly lower level compared to the parental strain (Figure 7b). At 96 h, all mutant strains except ∆*tryvc3* exhibited a reduction in CMCase activity when comparing the condition with 10 mM CaCl_2_ supplementation to the condition without the metal (Figure 7d). These findings suggest that the vacuolar transport proteins TrPMC1, TrYVC1, TrYVC3, and TrYVC4 play a crucial role in triggering endoglucanase production in response to calcium.

To understand how cellulase expressions are affected in the mutant strains, we analyzed the expression of the genes *cel7b* (endoglucanase I) (Figure 7e), *cel7a* (cellobiohydrolase I) (Figure 7f), *cel6a* (cellobiohydrolase II) (Figure 7g), and *cel3a* (beta-glucosidase I) (Figure 7h) in MA + 1% cellulose for 24 h, both with and without CaCl_2_ supplementation. Our results show that all these genes are induced in the parental strain when calcium is supplemented to the culture medium (Figure 7b). In contrast, in the mutant strains, although induction occurs, the levels are significantly lower or non-existent. For instance, the parental strain’s level of *cel7a* is approximately 512 times higher when 10 mM CaCl_2_ is added, while in ∆*trpmc1*, ∆*tryvc1*, ∆*tryvc3*, and ∆*tryvc4*, it is about 8, 6, 3, and 3 times higher, respectively (Figure 7f). These findings underscore that the calcium transport proteins characterized in this study are a key factor in cellulase production when calcium is supplemented to the medium. It is important to note that the MA medium already contains calcium at a concentration of 5.4 mM, so the supplementation resulted in a final concentration of 15.4 mM.

To gain a better understanding of the mechanism behind the observed cellulase inhibition in the mutant strains, we analyzed the expression of the genes encoding the transcription factors XYR1, ACE3, HAC1a, CRE1, and CRZ1, as well as the calcium signaling components CAM and CNA1 (Figure 8).

With respect to the transcription factors, we observed statistically significant differences only for *xyr1* and *hac1a* in the parental strain supplemented with CaCl_2_, compared to the control without supplementation (Figure 8a,c). This observation is consistent with the detected increase in cellulase expression. In the mutant strains, however, we did not observe an increase in *xyr1*, and for *hac1a*, an increase was only noted in the ∆*tryvc3* and ∆*tryvc4* strains, but to a lesser extent than in the parental strain (Figure 8a,c). Furthermore, we observed an increase in *crz1* expression in ∆*tryvc3* when supplemented with CaCl_2_ (Figure 8e), and a decrease in *ace3* in ∆*tryvc1* when supplemented with CaCl_2_ (Figure 8b), both when compared to the control condition without the addition of the metal.

Upon analyzing the expression of the components of the calcium-dependent signaling pathway, we observed an increase in *cam* expression in the parental strain and in ∆*tryvc4* (Figure 8f), as well as the upregulation of *cna1* in ∆*trpmc1*, and ∆*tryvc3* under calcium supplementation (Figure 8g). These results suggest that the production of cellulases is influenced by a combination of factors, including different transcription factors and the calcium signaling components.

### 3.4. The Production of Xylanases and Secondary Metabolites Are Impacted by the Vacuolar Calcium Transport Proteins in T. reesei in the Presence of Xylose

To comprehend the correlation between calcium supplementation and the observed growth enhancement in the parental strain in xylose, as well as the increased growth of the mutant strains relative to the parental strain without calcium supplementation, we cultivated these strains in liquid MA media with 25 mM xylose, both with and without calcium. We then analyzed the expression of genes encoding enzymes involved in xylose metabolism (XYL1, LAD1, and LXR3), xylanases (XYN1 and XYN2), and calcium signaling components (CAM, CNA1, and CRZ1) using the fungal mycelia.

Our analysis of the expression of genes encoding xylose metabolism enzymes revealed that *xyl1* and *lxr3* are more expressed in the parental strain than in the mutant strains without calcium supplementation (Figure 9a,c). Upon the addition of 10 mM CaCl_2_, we observed negative regulation of *xyl1* (Figure 9a) and positive regulation of *lad1* (Figure 9b) and *lxr3* (Figure 9c) in the parental strain, compared to the same strain without the metal. These genes may be induced in response to calcium supplementation, contributing to the enhanced growth observed in Figure 3. Despite the mutant strains showing increased growth without calcium supplementation, the only observed differences were the inhibition of *xyl1* in the ∆*trpmc1*, ∆*tryvc1,* and ∆*tryvc3* (Figure 9a), and *lxr3* in ∆*trpmc1* (Figure 9c), and a slight induction of *lad1* compared to the parental strain (Figure 9b).

As depicted in Figure 9d, we observed that the expression of *cam* was induced by calcium in both the parental and the mutant strains ∆*trpmc1* and ∆*tryvc1;* however, an increase in growth (Figure 3) was only evident in the parental strain. In the case of *cna1*, we noted an upregulation in the parental strain and in ∆*tryvc1* when calcium was supplemented to the culture medium (Figure 9e). Regarding the transcription factor CRZ1, its transcript levels decreased in the parental and ∆*tryvc4* strains upon calcium supplementation (Figure 9f). These findings suggest that the elevated levels of *cam* and *cna1*, observed in the absence of calcium supplementation, may contribute to the enhanced growth seen in the mutant strains. Furthermore, the lack of additional growth promotion upon calcium supplementation could be due to a compromised calcium signaling pathway.

Considering that xylanases are induced by media containing xylose [50], we investigated the effect of deleting vacuolar calcium transport proteins on their gene expression (Figure 9g,h). First, we observed that xylanase expression levels are elevated in the mutant strains compared to the parental strain. However, while calcium supplementation led to an increase in the expression of *xyn1* and *xyn2* in the parental strain, it resulted in inhibition in the mutant strains, except for *xyn1* in the ∆*tryvc1* strain. Notably, the repression of *xyn2* was significantly pronounced in the mutant strains compared to the same strains without calcium supplementation, suggesting that calcium-mediated signaling could play a crucial role in the expression of xylanases.

Interestingly, during the cultivation of the fungi for gene expression analysis, we observed that the addition of calcium resulted in the parental strain losing its characteristic yellow pigmentation (Appendix A). Consequently, we analyzed the expression of genes related to the production of the yellow pigment, a sorbicillinoid, produced by *T. reesei* [28]. Both the transcription factors *ypr1* (Figure 9j) and *ypr2* (Figure 9k), as well as the *sor1* gene (Figure 9i) involved in pigment production, are repressed when calcium is supplemented to the culture medium. This effect also occurs in the mutant strains ∆*trpmc1*, ∆*tryvc3*, and ∆*tryvc4*, albeit to a lesser extent, indicating that the production of this secondary metabolite is impaired when the calcium transport proteins are absent and when calcium is supplemented.

### 3.5. Microscopy Analysis Reveals the Role of TrPMC1, TrYVC1, TrYVC3, and TrYVC4 in Cell Wall Thickness and Calcium Dynamics in T. reesei

To gain deeper insights into the observed effects in mutant strains lacking vacuolar calcium transport proteins, we marked intracellular calcium using Fluo-4/AM in cultures with MA medium supplemented with 25 mM xylose and 10 mM CaCl_2_. As depicted in Figure 10, calcium appears more evenly distributed in the parental strain, without any distinct points indicating higher concentration, unlike in the mutant strains (Figure 10a). This suggests that in these strains, calcium may be accumulating in the vacuoles due to the deletion of the proteins involved in its transport.

Given our observation that the ∆*trpmc1*, ∆*tryvc1*, and ∆*tryvc3* mutant strains exhibit greater resistance to wall stress induced by CR than the parental strain (Figure 6a,c), we measured and calculated the wall thickness of the strains under study using calcofluor white labeling (CFW) (Figure 10b,c). Our analysis revealed that all mutant strains exhibited cell wall thickening when calcium was added (Figure 10c). However, in the absence of calcium, only the ∆*tryvc1* and ∆*tryvc3* strains were found to have thicker walls than the parental strain (Figure 10b). So, this result suggests the knockout strains have a high tolerance to stress agents, which could be explained by cell wall thickening.

## 4. Discussion

In this study, we functionally characterized four vacuolar calcium transport proteins in *T. reesei*: TrPMC1, TrYVC1, TrYVC3, and TrYVC4. First, we identified genes encoding proteins that are homologous to PMC1, YVC1, and VCX1 of *S. cerevisiae*, and then, we analyzed the expression profiles of these genes using various RNA-Seq datasets obtained by our group and constructed mutant lines with deletions of genes *58952*, *74057*, *55731*, and *56440*, which were subsequently named ∆*trpmc1*, ∆*tryvc1*, ∆*tryvc3*, and ∆*tryvc4*, respectively. We examined the growth phenotype of these strains, their cellulolytic enzyme activity, and the expression of genes associated with the degradation of lignocellulosic biomass, secondary metabolite biosynthesis, and the calcium signaling pathway. Finally, we conducted microscopy assays to confirm the intracellular localization of calcium in the parental and mutant strains of *T. reesei*.

Through bioinformatics analyses, we identified 5 potential YVC1, 7 potential VCX1, and 21 potential PMC1 proteins (Table 1). However, only 15 of these were identified as proteins that transport calcium in the vacuoles (5 YVC1, 7 VCX1, and 3 PMC1), with the others likely involved in the transport of phospholipids, calcium, and other metals in other organelles. We also identified proteins of the SERCA (sarcoplasmic/endoplasmic reticulum Ca^2+^ ATPase) and PMCA (plasma membrane Ca^2+^ ATPase) types, as proteins present in other cellular compartments may share similarities with vacuolar transport proteins [15]. It is important to note that proteins classified as PMC1 can be located both in the plasma membrane and in the vacuoles, as is the case with the Ca^2+^ ATPases NCA-2 and NCA-3 from *Neurospora crassa* [51]. A similar number of transport proteins involved in calcium signaling was reported by [16], who identified three ion channels, 10 ATPases, and 10 calcium transporters in *T. reesei*. Additionally, other organisms share a similar number of YVC1, VCX1, and PMC1 homologs. For instance, four homologs of YVC1 were identified in *Colletotrichum graminicola* (CgTRPF1-4) [52], three homologs of PMC1 were identified in *Aspergillus fumigatus* (PMCAA-C) [53] and in *Beauveria bassiana* (PMCA-C) [54], and five VCX1 homologs were also identified in *B. bassiana* (VCX1A-E) [55].

When analyzing the expression profiles of proteins identified in the RNA-Seq data obtained by our group [17,18,41,48,56], we observed that expression levels vary even between the strains that are generally used as parental in functional studies (Figure 2): QM6a, QM9414, and TU6 [57]. This indicates that the mutations present in QM9414 and TU6 strains, which result in greater cellulase production, may be involved in the calcium-mediated signaling pathway. Furthermore, we observed that the expression of the genes under study can be regulated by the transcription factors XYR1, CRE1, and AZF1, and are also involved with the MAPK (mitogen-activated protein kinase) signaling pathway, as these proteins have also been differentially expressed in these analyses. Consistent with these data, it has been demonstrated that calmodulin, an important component of the calcium-mediated signaling pathway, binds to an integral membrane protein, activating the MAPK signaling pathway in *Candida albicans* [58]. Furthermore, in *B. bassiana*, it was demonstrated that the inactivation of Slt2/Mpk1 repressed the phosphatase activity of calcineurin [38]. The carbon source can also influence the expression of the proteins under study. For example, in *S. cerevisiae* [59] it was demonstrated that the addition of glucose causes the entry of calcium into the cell and that this process may be related to the YVC1 vacuolar channel, as its deletion resulted in a strain in which this response was attenuated.

Based on the collected data, the proteins identified by JGI IDs 58952, 74057, 55731, and 56440 were selected for further investigation. The objective was to generate mutant strains with the deletion of the respective genes, thereby facilitating a more comprehensive understanding of the calcium-dependent signaling pathway. These proteins were subsequently designated as TrPMC1, TrYVC1, TrYVC3, and TrYVC4, respectively.

Our phenotypic growth analyses revealed an enhanced growth rate in the presence of arabinose, xylitol, and xylose in the mutant strains compared to the parental strain (Figure 3). While previous studies have demonstrated that calcium can enhance fungal growth in media containing glucose [10], there is limited evidence regarding its effect in media containing xylose. In *Mucor circinelloides*, it has been shown that zinc and calcium can augment xylose consumption. The underlying mechanism remains elusive, but it is hypothesized that the concentration of calcium may regulate zinc transport into the cell [60]. In this study, we examined the expression of genes involved in xylose metabolism and calcium signaling in both parental and mutant strains cultured in xylose (Figure 9). It is plausible that the inhibition of *xyl1* and *lxr3*, coupled with the induction of *lad1*, may play a role in this process. This could be accompanied by an upregulation of *cam* and *cna1*, although the precise mechanism requires further investigation.

We also demonstrated that calcium influences the production of secondary metabolites and that vacuolar calcium transport proteins may be implicated in this process (Figure 9). This observation aligns with findings in *B. bassiana*, where *cam* plays a significant role in secondary metabolism by interacting with a ketoisovalerate reductase, thereby inhibiting its activity [61]. These findings are consistent with a study conducted in *Penicillium oxalicum*, which showed that PoCrz1 regulates secondary metabolism. In the absence of this transcription factor, five clusters were negatively regulated in the mutant strain [62]. Furthermore, it has been shown that intracellular calcium is involved in the expression of genes related to the biosynthesis and secretion of metabolites in filamentous fungi, with YVC1 playing a role. The exact mechanism remains unknown, but it is suggested that calcium signaling may induce the expression of the regulator *crc* (a zinc-finger type regulator), which is responsible for the induction of secondary metabolite biosynthesis genes [63]. Additionally, it has been demonstrated that PenV, a putative calcium channel in *Penicillium chrysogenum*, plays a crucial role in secondary metabolite production. By transporting amino acids from the vacuoles to the cytosol, PenV supports the biosynthesis of penicillin. Furthermore, PenV regulates the expression of two penicillin biosynthesis genes, pcbC, and penDE, highlighting its significance in secondary metabolite production [63,64].

In xylose, we observed that the mutant strains exhibit sensitivity to high concentrations of manganese, and interestingly, the growth enhancement typically induced by calcium in the parental strain was absent in these mutants (Figure 4). This aligns with previous findings indicating that manganese plays a role in calcium signaling in *T. reesei*, as it triggers an increase in cytosolic calcium concentration [7]. Furthermore, it has been shown that YVC1 can be activated by calcium or other metals, such as manganese, magnesium, and zinc [15], thereby facilitating the release of calcium from the vacuoles. In addition, when the strains were cultivated under osmotic stress conditions, we found that calcium supplementation hindered growth in the mutant strains, but not in the parental strain, within the first 48 h (Figure 5). This is consistent with several studies indicating that calcium signaling contributes to osmotic stress tolerance. For instance, in *B. bassiana* and *N. crassa*, disruption of calcineurin function resulted in increased sensitivity to osmotic stress [65,66]. Additionally, it has been shown that phospholipase C may also play a role in osmotic stress resistance in *S. cerevisiae* [66].

Here, we also observed that the mutant strains exhibited increased resistance to cell wall stress induced by CR (Figure 6). Using confocal microscopy, we found that the cell wall of *T. reesei* is thicker in mutant strains (Figure 10), which could explain their heightened resistance to the cell wall stress caused by CR. In *Aspergillus nidulans*, it was shown that deletion of CchA, a plasma membrane calcium channel of the high-affinity calcium influx system (HACS) and MidA, its regulatory unit, rendered the mutant strains more resistant to stress induced by CR and CFW, likely due to alterations in the composition of the fungal cell wall [67]. In fact, the transcription factor CRZ1/CRZA is involved in cell wall biosynthesis by the induction of the expression of the genes *chs1*, *crh1*, *rho1*, *scw10*, and *kre6* in *Saccharomyces cerevisiae* [68], chitin synthase (*chs6*) in *Cryptococcus neoformans* [69] and *chsA-G* in *A. fumigatus* [70]. In this study, we found that in the presence of xylose and with the supplementation with 10 mM CaCl2, the mutant strains show induced Crz1 in relation to the parental strain, which could explain the resistance that was observed in the presence of the metal, this suggesting that the vacuolar transport proteins TrPMC1, TrYVC1, TrYVC3 and TrYVC4 might be involved in the cell wall resistance though CRZ1.

Our results also showed that the growth of mutant strains is compromised when cultivated in CMC (Figure 6), and that the expression and activity of cellulases are significantly reduced in these strains (Figure 7). While calcium supplementation is known to enhance cellulase production in *T. reesei* [10], it had a negative impact on the mutant strains. This suggests that the uptake and release of calcium from the vacuoles are crucial for the induction of cellulase expression in *T. reesei*. Other studies have also linked calcium signaling to cellulase production in *T. reesei* [7,9,11,22,23], but this is the first to demonstrate that vacuolar calcium transport proteins are also involved in this process. We noted the downregulation of the inducers *ace3* and *crz1* in the TrYVC1-deficient mutant strain and the upregulation of the repressor *cre1* in the TrYVC3-deficient strain both in the presence of calcium (Figure 8).

Given the lack of a clear pattern, it is possible that the reduced production of cellulases in the mutant strains could be associated with other proteins. In addition, despite the involvement of YVC1 deletion mutants in the release of calcium and the PMC1 mutant in the sequestration of calcium, we did not observe distinct phenotypic differences. This could be due to the presence of other homologs in *T. reesei*. However, we found that the deletion of a single gene is sufficient to impair cellulase production (Figure 7). It is possible that the observed effects are due to the retention of calcium within the vacuoles, as shown using confocal microscopy (Figure 10). It is conceivable that this retention hampers calcium-dependent signaling, which has been shown to be crucial for the induction of cellulases [7]. This result is in accordance with the study of Martins-Santana et al. (2020) [11], in which they showed that the promoter regions of cellulase genes, as well as calcium transport-related genes, have binding sites for the regulator CRZ1. In addition, other calcium signaling components were shown to be involved in cellulases production. For example, *cam* and *cna1* are upregulated when manganese is added to the culture medium, a condition in which endoglucanases and cellobiohydrolases production is induced [7]. Altogether, the data presented here suggests that the calcium signaling pathway is crucial to cellulases production in *T. reesei*. 

## 5. Conclusions

In conclusion, our study provides novel insights into the role of calcium signaling and vacuolar calcium transport proteins in the regulation of growth, stress response, secondary metabolism biosynthesis, and cellulase production in *T. reesei* (Figure 11). We demonstrated that the deletion of specific genes involved in calcium transport can significantly impact these biological processes, suggesting their potential as targets for genetic manipulation to enhance the industrial utility of this organism. However, the exact mechanisms underlying these effects remain to be elucidated, highlighting the need for further research in this area. Our findings pave the way for future studies exploring the intricate network of calcium signaling in fungi and its implications for industrial biotechnology.

## Figures and Tables

**Figure 1 jof-10-00853-f001:**
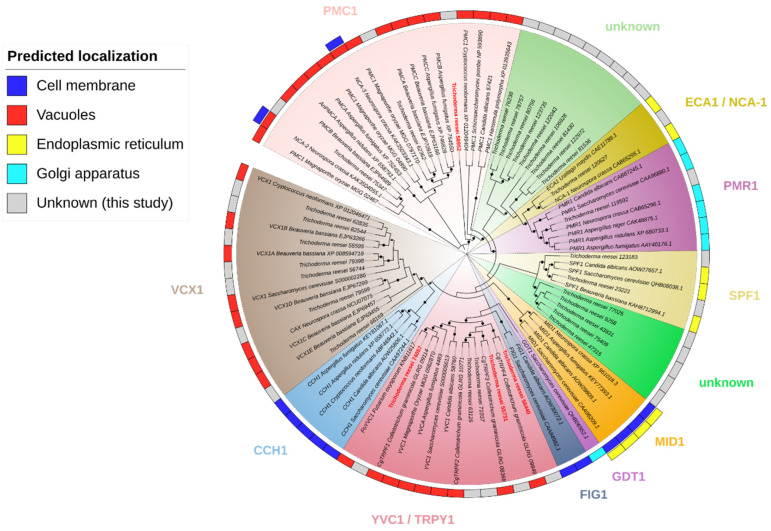
Cladogram of calcium transport proteins. We used sequences of characterized proteins present in the cell membrane, vacuoles, endoplasmic reticulum, and Golgi apparatus together with *T. reesei* proteins of unknown localization identified in this study. The support values from 1000 resamples are indicated by the black circles, varying from 0 to 1. The figure only shows values from 0.8 to 1, from smaller to larger size.

**Figure 2 jof-10-00853-f002:**
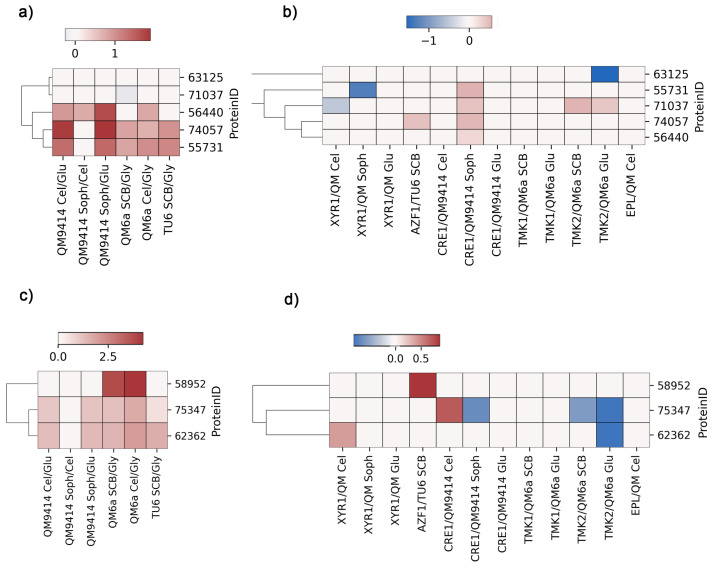
Comparative expression analysis of *T. reesei* putative vacuolar calcium transport proteins, homologous to *S. cerevisiae’*s YVC1 and PMC1. The RNA-Seq data were obtained from studies [6,17,18,19,20,48]. (**a**) *yvc1* expression in commonly used parental strains and (**b**) the mutant strains lacking *Xyr1*, *Azf1*, *Cre1*, *Tmk1*, *Tmk2* and *Epl2*, respectively. (**c**,**d**) *pmc1* expression under the same conditions. Cel = cellulose, Glu = glucose, Soph = sophorose, SCB = sugarcane bagasse, Gly = glycerol.

**Figure 3 jof-10-00853-f003:**
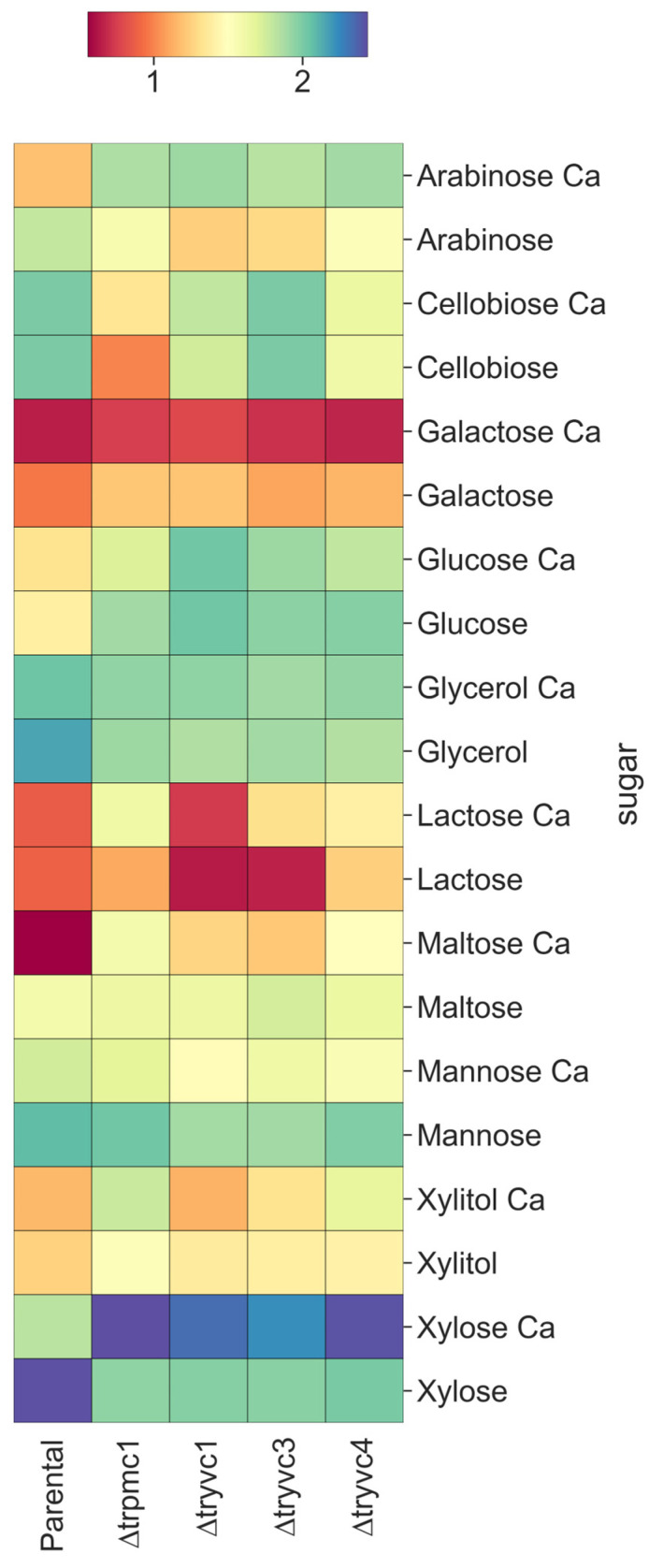
Growth of strains QM6a∆*tmus53*∆*pyr4* (parental), ∆*trpmc1*, ∆*tryvc1*, ∆*tryvc3*, and ∆*tryvc*4 in minimal media in the presence of 25 mM of different carbon sources for 72 h, with and without 10 mM CaCl_2_ supplementation. The values represent the absorbance readings at 750 nm.

**Figure 4 jof-10-00853-f004:**
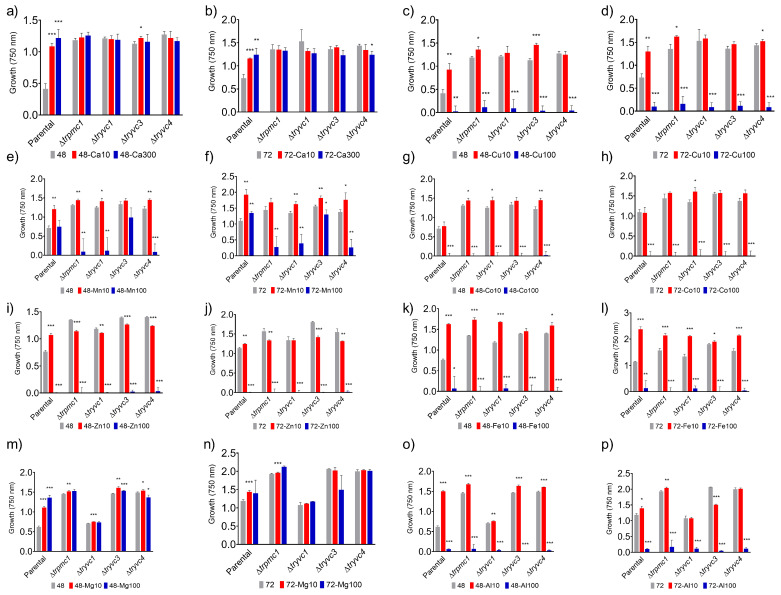
Growth of strains QM6a∆*tmus53*∆*pyr4* (parental), ∆*trpmc1*, ∆*tryvc1*, ∆*tryvc3*, and ∆*tryvc*4 in 25 mM xylose minimal media for 48 or 72 h in the presence of 10, 100, or 300 mM of the metals calcium (**a**,**b**), copper (**c**,**d**), manganese (**e**,**f**), cobalt (**g**,**h**), zinc (**i**,**j**), iron (**k**,**l**), magnesium (**m**,**n**) or aluminum (**o**,**p**). Significance levels are indicated as follows: * *p* < 0.05, ** *p* < 0.01, and *** *p* < 0.001 in relation to the parental strain (Student’s *t*-test). The control condition (gray bars) was calculated for each microplate.

**Figure 5 jof-10-00853-f005:**
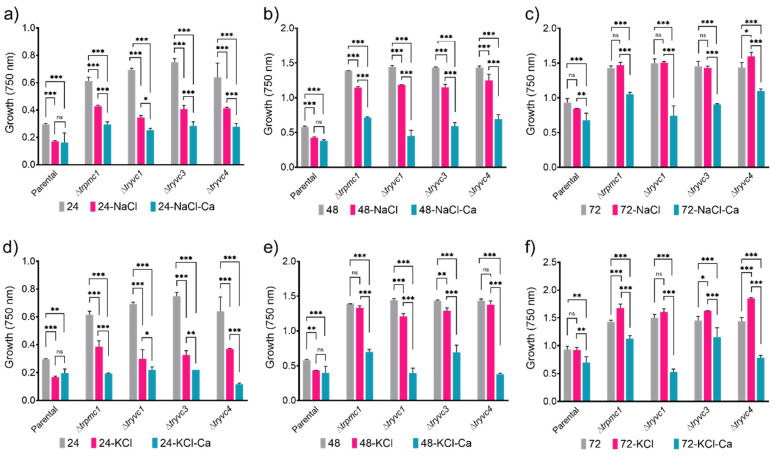
Growth of strains QM6a∆*tmus53*∆*pyr4* (parental), ∆*trpmc1*, ∆*tryvc1*, ∆*tryvc3*, and ∆*tryvc*4 in 25 mM xylose Minimal media in the presence of 0.5 M NaCl (**a**–**c**) or 0.5 M KCl (**d**–**f**) and in presence or absence of 10 mM CaCl_2_. * *p* < 0.05, ** *p* < 0.01, and *** *p* < 0.001 in relation to the parental strain (two-way ANOVA, followed by Tukey’s multiple comparisons). Non-significant results are indicated as ‘ns’. The control condition (gray bars) was calculated for each microplate.

**Figure 6 jof-10-00853-f006:**
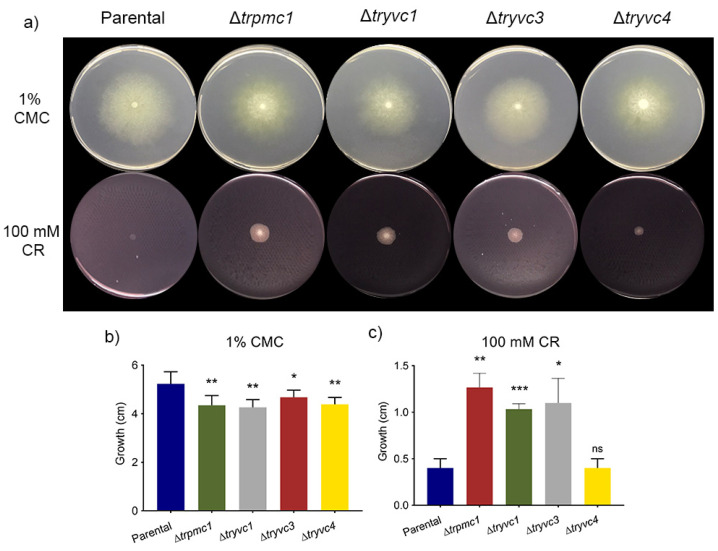
Growth analysis of strains QM6a∆*tmus53*∆*pyr4* (parental), ∆*trpmc1*, ∆*tryvc1*, ∆*tryvc3*, and ∆*tryvc*4 in MA with 1% carboxymethylcellulose (CMC) for 5 days or 2% glucose + 100 mM Congo red (CR) for 3 days. (**a**) growth in plates, (**b**) growth measurements in 1% CMC and (**c**) 100 mM CR. Significance levels are indicated as follows: * *p* < 0.05, ** *p* < 0.01, and *** *p* < 0.001 (Student’s *t*-test). Non-significant results are indicated as ‘ns’.

**Figure 7 jof-10-00853-f007:**
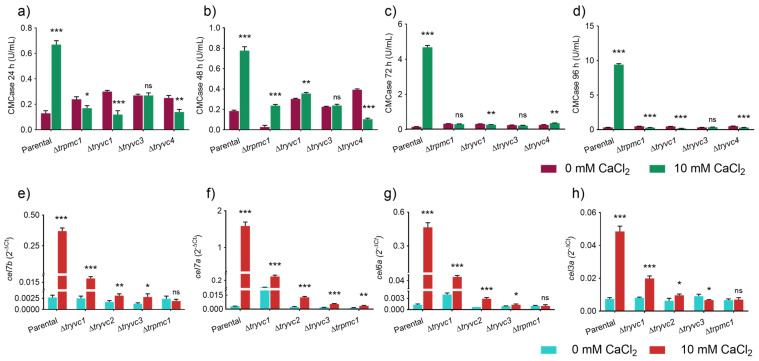
CMCase activity of parental and mutant strains grown for 24 h in MA media with 1% cellulose with or without 10 mM CaCl_2_ supplementation for 24 h (**a**), 48 h (**b**), 72 h (**c**), and 96 h (**d**), and gene expression of endoglucanase (*cel7b*) (**e**), cellobiohydrolases (*cel7a* and *cel6a*) (**f**,**g**) and beta-glucosidase (*cel3a*) (**h**). Significance levels are indicated as follows: * *p* < 0.05, ** *p* < 0.01, and *** *p* < 0.001 (Student’s *t*-test). Non-significant results are indicated as ‘ns’.

**Figure 8 jof-10-00853-f008:**
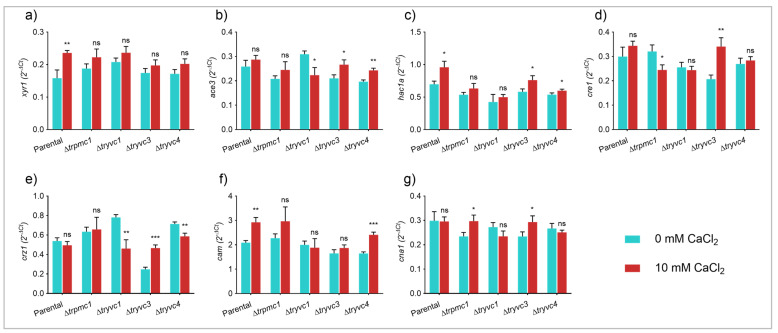
Gene expression of transcription factors and calcium signaling components of parental and mutant strains grown on MA media with 1% cellulose for 24 h. Gene expression of *xyr1* (**a**), *ace3* (**b**), *hac1a* (**c**), *cre1* (**d**), *crz1* (**e**), *cam* (**f**), and *cna1* (**g**). * *p* < 0.05, ** *p* < 0.01, and *** *p* < 0.001 (Student’s *t*-test). Non-significant results are indicated as ‘ns’.

**Figure 9 jof-10-00853-f009:**
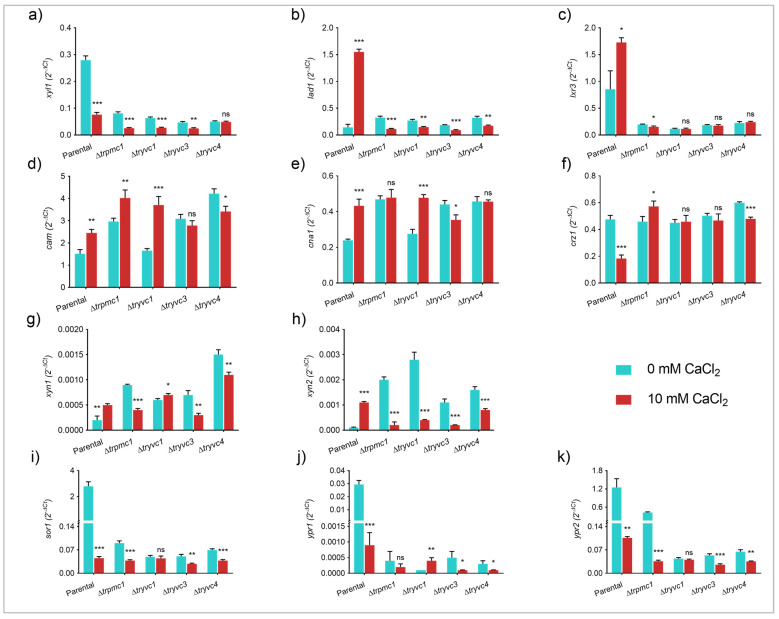
Gene expression of xylose metabolism enzymes (**a**–**c**), calcium signaling components (**d**–**f**), xylanases (**g**,**h**), and secondary metabolites components (**i**–**k**) of parental and mutant strains grown on MA media with 25 mM xylose for 48 h. * *p* < 0.05, ** *p* < 0.01, and *** *p* < 0.001 (Student’s *t*-test). Non-significant results are indicated as ‘ns’.

**Figure 10 jof-10-00853-f010:**
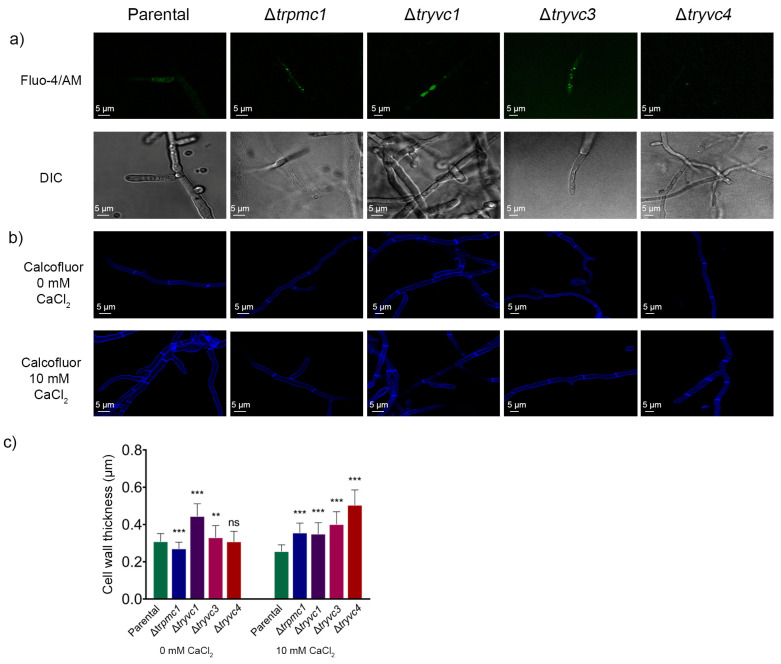
Confocal microscopy and analysis of cell wall thickness. (**a**) the strains were grown in MA media with 25 mM xylose supplemented with 10 mM CaCl_2_ and stained with 5 µM Fluo-4/AM. (**b**) the strains were grown in MEX media supplemented with 0 or 10 mM CaCl_2_ for 1 day in a microscope slide and stained with 0.001% calcofluor white. (**c**) measurements of cell wall thickness. ** *p* < 0.01 and *** *p* < 0.001 (Student’s *t*-test). Non-significant results are indicated as ‘ns’.

**Figure 11 jof-10-00853-f011:**
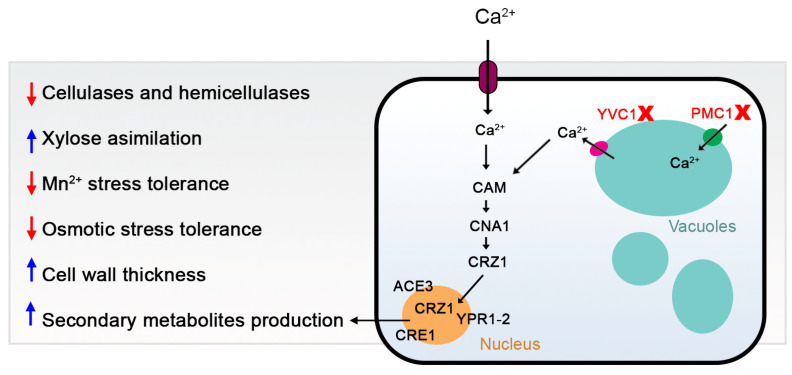
Putative model of the function of the vacuolar calcium transport proteins YVC1 and PMC1 of *T. reesei*. The deletion of these proteins impairs the dynamics of calcium transport in the vacuoles, therefore, when extracellular calcium enters the cells, the calcium-mediated signaling does not occur properly, as the expression of the transcription factors ACE3, CRZ1, CRE1, YPR1, and YPR2 are increased/decreased, causing a reduction in cellulases/hemicellulases expression, manganese and osmotic stress tolerance and an increase in xylose assimilation and in cell wall thickness. red arrows down—down-regulation effect and blue arrows up—up-regulation effect.

**Table 1 jof-10-00853-t001:** Potential vacuolar calcium transporters and channels in *T. reesei*. The proteins under study are highlighted in bold and underlined.

JGI ID	Possible Homologue in *S. cerevisiae*	Size (aa)	Annotation	Full e-Value (Hmmsearch)
74057	YVC1	625	Nonselective cation channel protein	1.8 × 10^−256^
55731	YVC1	1167	Nonselective cation channel protein	7 × 10^−164^
56440	YVC1	701	Nonselective cation channel protein	5 × 10^−145^
63125	YVC1	631	Nonselective cation channel protein	3.7 × 10^−121^
71037	YVC1	1104	Nonselective cation channel protein	3.4 × 10^−117^
79599	VCX1	462	Vacuolar calcium ion transporter	1.4 × 10^−178^
55595	VCX1	421	Vacuolar calcium ion transporter	2 × 10^−166^
79398	VCX1	339	Vacuolar calcium ion transporter	2.1 × 10^−151^
56744	VCX1	381	Vacuolar calcium ion transporter	8.5 × 10^−100^
82544	VCX1	433	Vacuolar calcium ion transporter	4.3 × 10^−87^
68169	VCX1	394	Vacuolar calcium ion transporter	1 × 10^−81^
62835	VCX1	1115	Vacuolar calcium ion transporter	7.4 × 10^−48^
75347	PMC1	1379	Calcium-translocating P-type ATPase, PMCA-type	0
58952	PMC1	1204	Calcium-translocating P-type ATPase, PMCA-type	0
62362	PMC1	1281	Calcium-translocating P-type ATPase, PMCA-type	0
120627	PMC1	998	Calcium-translocating P-type ATPase, SERCA-type	2.4 × 10^−142^
119592	PMC1	1062	Calcium-transporting ATPase	1.5 × 10^−137^
81536	PMC1	1101	Calcium-transporting ATPase	4.3 × 10^−111^
122972	PMC1	1071	Calcium-transporting ATPase	8.4 × 10^−109^
81430	PMC1	1023	Calcium-transporting ATPase	1.3 × 10^−106^
106928	PMC1	1049	Sodium/potassium-transporting ATPase subunit alpha	6.3 × 10^−94^
78757	PMC1	923	Calcium-transporting ATPase	1.9 × 10^−47^
76238	PMC1	982	Calcium-transporting ATPase	1.4 × 10^−45^
123183	PMC1	1309	Cation-transporting ATPase-related	1.1 × 10^−38^
122043	PMC1	1171	Heavy metal translocating P-type ATPase	2.2 × 10^−36^
43831	PMC1	1354	Probable phospholipid-transporting ATPase	1.2 × 10^−32^
23221	PMC1	1318	Cation-transporting ATPase-related	3.6 × 10^−31^
123735	PMC1	1105	Heavy metal translocating P-type ATPase	5 × 10^−30^
77025	PMC1	1300	Probable phospholipid-transporting ATPase	7.4 × 10^−30^
80756	PMC1	1133	Heavy metal translocating P-type ATPase	1 × 10^−27^
79258	PMC1	1534	Probable phospholipid-transporting ATPase	3.2 × 10^−27^
47315	PMC1	1392	Probable phospholipid-transporting ATPase	2.7 × 10^−23^
75409	PMC1	1368	Probable phospholipid-transporting ATPase	2.4 × 10^−20^

## Data Availability

The original contributions presented in the study are included in the article/Appendix A, further inquiries can be directed to the corresponding author.

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
