# Peer review of "Regulatory Role of Vacuolar Calcium Transport Proteins in Growth, Calcium Signaling, and Cellulase Production in Trichoderma reesei"

_jof, 2024, doi:10.3390/jof10120853_

Round 1

Reviewer 1 Report

This manuscript investigates comprehensively the roles of four vacuolar calcium transport proteins in growth, calcium signaling, and cellulase production in Trichoderma reesei, by gene knockout and sorts of phenotype analysis. However, I can not find the Supplementary Materials. The authors should submit that.

1. Using absorbance to determine the filamentous fungal growth, what about the precision? The authors should cite referance here.

2. As for carboxymethylcellulose (CMC), the meaning of the abbreviation should be marked when it first appears. for Figure 6. please indicate the meaning of the abbreviation CMC in the figure legend.

3. In Figure 10, μM is used to represent the thickness unit, which seems inappropriate and should be μm.

4. “3.1. Identification of putative homologues in Trichoderma reesei: 3 for PMC1, 5 for YVC1, and 7 for VCX1 of Saccharomyces cerevisiae ”, here is not the first appearance of these two species , and T. reesei and S. cerevisiae should be used respectively.

Author Response

Response to Reviewer 1 Comments

1. Summary

2. Point-by-point response to Comments and Suggestions for Authors

Major comments

This manuscript investigates comprehensively the roles of four vacuolar calcium transport proteins in growth, calcium signaling, and cellulase production in Trichoderma reesei, by gene knockout and sorts of phenotype analysis. However, I can not find the Supplementary Materials. The authors should submit that.

Response: Thank you for your comments. We have submitted the Supplementary Materials as a separate file.

Comment 1: Using absorbance to determine the filamentous fungal growth, what about the precision? The authors should cite reference here.

Response 1: Thank you for pointing this out. While we acknowledge the potential limitations of this method, we believe that in our specific experimental context, turbidity measurements provide a reliable and informative assessment of fungal growth. This is supported by the consistent correlation between turbidity and radial growth observed in both microplate and petri dishes experiments. For instance, the decreased growth of mutant strains in the presence of cellobiose was observed in both experimental setups. We agree that we should provide the reference, therefore we have updated the methods section with more details about the test and the reference. The revisions are marked in red in the manuscript (page 3, lines 143-146).

Comment 2: As for carboxymethylcellulose (CMC), the meaning of the abbreviation should be marked when it first appears. for Figure 6. please indicate the meaning of the abbreviation CMC in the figure legend.

Comment 4: “3.1. Identification of putative homologues in Trichoderma reesei: 3 for PMC1, 5 for YVC1, and 7 for VCX1 of Saccharomyces cerevisiae”, here is not the first appearance of these two species, and T. reesei and S. cerevisiae should be used respectively.

Response 4: Thank you for pointing this out. We have modified it in the manuscript (page 6, lines 294 and 295).

Reviewer 2 Report

Regulatory Role of Vacuolar Calcium Transport Proteins in Growth, Calcium Signaling, and Cellulase Production in Trichoderma reesei

Introduction section

The Introduction section of the paper is quite comprehensive and well-structured, providing a good background for the research.

Materials and Methods section

All abbreviations mentioned in the text should be explained (Line 137, CMC). Actually, the abbreviation is explained later (Line 225 and 415) and should be explained when it is first mentioned in the text.

The text mentions Figures (Lines 170, 180, 214, 356, 362, 364, 533) and Tables (Lines 178, 197, 200, 215, 260, 275, 292) which are part of the supplementary material, and this material we not submitted for review.

I believe that the statistical analysis part needs to be explained in more detail. Which programs, tests were used... Tests are mentioned later in the text, which should also be listed here.

Results section

In general, it is difficult to follow the names of mutant strains and genes in the text, ie. their markings. However, this part of the paper is concise and well written.

Discussion section

I would like to praise the authors for the very graphic presentation of the entire paper in Figure 11. This picture can also be a graphic abstract of this work.

The references include many works published during the 20th century. I believe that some of them can probably be replaced by some newer references.

General opinion

The results highlight the central role of calcium signaling in fungal physiology, particularly in growth, stress response, and enzyme production. Also, I see that the authors have published many papers with T. reesei. Given the role of T. reesei in industrial cellulase production, understanding the regulatory role of vacuolar calcium transport proteins has practical importance.

I suggest that the paper be considered for publication after minor revisions.

Best regards,

Reviewer

Regulatory Role of Vacuolar Calcium Transport Proteins in Growth, Calcium Signaling, and Cellulase Production in Trichoderma reesei

Materials and Methods section

All abbreviations mentioned in the text should be explained (Line 137, CMC). Actually, the abbreviation is explained later (Line 225 and 415) and should be explained when it is first mentioned in the text.

The text mentions Figures (Lines 170, 180, 214, 356, 362, 364, 533) and Tables (Lines 178, 197, 200, 215, 260, 275, 292) which are part of the supplementary material, and this material we not submitted for review.

I believe that the statistical analysis part needs to be explained in more detail. Which programs, tests were used... Tests are mentioned later in the text, which should also be listed here.

Best regards,

Reviewer

Author Response

 1. Summary 
Thank you very much for taking the time to review this manuscript. Please find the detailed responses below and the corresponding revisions/corrections highlighted/in track changes in the re-submitted files.

Materials and Methods section 

All abbreviations mentioned in the text should be explained (Line 137, CMC). Actually, the abbreviation is explained later (Line 225 and 415) and should be explained when it is first mentioned in the text.

The text mentions Figures (Lines 170, 180, 214, 356, 362, 364, 533) and Tables (Lines 178, 197, 200, 215, 260, 275, 292) which are part of the supplementary material, and this material we not submitted for review.

I believe that the statistical analysis part needs to be explained in more detail. Which programs, tests were used... Tests are mentioned later in the text, which should also be listed here.

Results section 

In general, it is difficult to follow the names of mutant strains and genes in the text, ie. their markings. However, this part of the paper is concise and well written.

Discussion section 

I would like to praise the authors for the very graphic presentation of the entire paper in Figure 11. This picture can also be a graphic abstract of this work.

The references include many works published during the 20th century. I believe that some of them can probably be replaced by some newer references.

General opinion 

The results highlight the central role of calcium signaling in fungal physiology, particularly in growth, stress response, and enzyme production. Also, I see that the authors have published many papers with T. reesei. Given the role of T. reesei in industrial cellulase production, understanding the regulatory role of vacuolar calcium transport proteins has practical importance.

I suggest that the paper be considered for publication after minor revisions.

Best regards,

Reviewer

Response: Thank you for your insightful comments. We appreciate your thorough review of our manuscript. In response to your comments, we have made the following changes:

We have defined "CMC" on its first appearance in the text (line 137).

We have included Supplementary Materials at the end of the manuscript (line 971).

We have provided additional details about the statistical analysis used in the study (lines 288-291 and line 413).

We have removed some 20th century references and replaced some with more recent publications (lines 183, 193, 196, 205, 207, 814, 815, 838, 839).

Round 2

Reviewer 1 Report

The authors has revised the manuscript carefully accoding to the previous comments and reached the publishable level.

The authors has revised the manuscript carefully accoding to the previous comments and reached the publishable level. There are no detail comments.